# Impacts of Radio-Frequency Electromagnetic Field (RF-EMF) on Lettuce (*Lactuca sativa*)—Evidence for RF-EMF Interference with Plant Stress Responses

**DOI:** 10.3390/plants12051082

**Published:** 2023-02-28

**Authors:** Nam Trung Tran, Luca Jokic, Julian Keller, Jens Uwe Geier, Ralf Kaldenhoff

**Affiliations:** 1Applied Plant Sciences, Department of Biology, Technical University Darmstadt, 64287 Darmstadt, Germany; 2Forschungsring e.V., 64295 Darmstadt, Germany

**Keywords:** radio frequency electromagnetic fields, plants, lettuce, photosynthesis, flowering, stress

## Abstract

The increased use of wireless technology causes a significant exposure increase for all living organisms to radio frequency electromagnetic fields (RF-EMF). This comprises bacteria, animals, and also plants. Unfortunately, our understanding of how RF-EMF influences plants and plant physiology remains inadequate. In this study, we examined the effects of RF-EMF radiation on lettuce plants (*Lactuca sativa*) in both indoor and outdoor environments using the frequency ranges of 1890–1900 MHz (DECT) at 2.4 GHz and 5 GHz (Wi-Fi). Under greenhouse conditions, RF-EMF exposure had only a minor impact on fast chlorophyll fluorescence kinetics and no effect on plant flowering time. In contrast, lettuce plants exposed to RF-EMF in the field showed a significant and systemic decrease in photosynthetic efficiency and accelerated flowering time compared to the control groups. Gene expression analysis revealed significant down-regulation of two stress-related genes in RF-EMF-exposed plants: violaxanthin de-epoxidase (VDE) and zeaxanthin epoxidase (ZEP). RF-EMF-exposed plants had lower Photosystem II’s maximal photochemical quantum yield (F_V_/F_M_) and non-photochemical quenching (NPQ) than control plants under light stress conditions. In summary, our results imply that RF-EMF might interfere with plant stress responses and reduced plant stress tolerance.

## 1. Introduction

Wireless communication was invented at the end of the nineteenth century, yet it has quickly become recognized as a significant innovation. It is hard to envisage our society without the billions of wireless-connected cell phones, tablets, and computers. These technologies have become indispensable partners in our big and small everyday activities. Inevitably, we are witnessing a massive increase in the exposure of animals and plants to man-made radio frequency electromagnetic fields (RF-EMF). In addition to their intensity, the characteristics of anthropogenic RF-EMF are significantly different to natural RF-EMF discharges (such as during a thunderstorm [1], Table 1). Therefore, it could be stated that the extraordinary level of RF-EMF we are presently exposed to is a unique phenomenon in Earth’s history. Certainly, most organisms are evolutionarily not prepared for it. As operators of wireless devices, humans are repeatedly subjected to RF-EMF exposure at very close distances. Not surprisingly, most research on the biological effects of RF-EMF was carried out on humans and vertebrate model animals [2]. While it is generally accepted that RF-EMFs cause a wide range of adverse effects on humans and could even be considered a “Possible Human Carcinogen”, the magnitude and severity of the risk remain disputed [3,4,5]. Several studies have demonstrated that RF-EMFs are only harmful when they reach certain thresholds, and that exposure at lower levels is generally harmless [6,7]. Based on these findings, many nations, notably the European Union and the United States, limit the maximum allowable RF-EMF exposure levels in public spaces [8,9].

In comparison to humans and animals, research on the impacts of RF-EMF on plants is considerably scarcer [10]. RF-EMF research on plants demands more attention because of several reasons. Firstly, plants can be more susceptible to RF-EMF radiation than animals. Lacking mobility, plants within an irradiation covered area are constantly exposed to RF-EMF. Furthermore, the very high surface area to weight ratios characteristic for most plants results in larger radiation absorption [11]. Secondly, the effects of RF-EMF on plants should be considered in light of their critical functions in the biosphere. Plants play an essential role as food sources for humans and animals. They are also essential components of many biogeochemical cycles and numerous other metabolic processes. Therefore, even minor impacts on plants can have significant and unpredictable secondary effects on human existence and generally, on ecosystems.

The effects of RF-EMF on plants are complex [2]. Two important homeostatic systems, calcium movement and reactive oxygen species (ROS) formation, are both affected by RF-EMF. Tomato plants (*Lycopersicon esculentum*) subjected to 900 MHz irradiation showed increased levels of calmodulin *calm-n6* and the calcium-dependent protein kinase *lecdpk1* [12]. After being exposed to 900 MHz radiation, increased H_2_O_2_ accumulation and significant upregulation of ROS scavenging enzyme activities were detected in mung beans (*Vigna radiata*) [13]. Significantly higher number of mitotic abnormalities were detected in onion cells (*Allium cepa*) after treatment with 400 MHz EMF compared to control samples [14]. RF-EMF also causes numerous changes in gene expressions. In tomato (*Lycopersicon esculentum*), at least five stress-related genes are rapidly upregulated in response to RF-EMF exposure: transcription factor *lebzip1*, calmodulin *calm-n6*, calcium-dependent protein kinase *lecdpk*, chloroplast mRNA binding protein *cmbp* and proteinase inhibitor-2 *pin2* [12,15,16,17]. In tomato fruits, irradiation with 9.3 GHz causes a reduction in the expression of ethylene-related genes amino cyclopropane-1-carboxylic acid (ACC) synthase and ACC oxidase [18]. A large-scale proteomic analysis in RF-EMF-exposed sunflower (*Helianthus annuus*) revealed modified abundance of almost 100 proteins [19]. In contrast, microarray analysis using cell suspension cultures of *Arabidopsis thaliana* detected only five genes whose expression levels were significantly different from those of control samples [20].

Exposure to RF-EMF has also been demonstrated to influence plant growth, development, photosynthesis, stress alleviation, and other metabolic activities. Exposure to 156–162 MHz caused duckweed (*Spirodela polyrhiza*) to grow at a lower rate and induced morphological and developmental abnormalities [21]. Reduced growth rates were also observed with irradiated mung bean (*Vigna radiata*) [13], bush rose (*Rosa hybrida*) [22], and radish (*Raphanus sativus*) [23]. RF-EMF alters cellular metabolism. RF-EMF exposure increases the enzymatic activity of several enzymes: catalase, ascorbate peroxidase [24], isocitrate dehydrogenase, malate dehydrogenase, glucose-6-phosphate dehydrogenase [25], α-and β-amylases [26], glutathione reductase, peroxidase, nitric oxide synthase, and superoxide dismutase [27]. In other research, a reduction in enzymatic activities has been observed in irradiated plants: acid and alkaline phosphatases [26], starch phosphorylase [27], polygalacturonase, pectinmethylesterase and β-galactosidase [18]. Parsley (*Petroselinum crispum*), dill (*Anethum graveolens*), and celery (*Apium graveolens*) exposed to RF-EMF displayed decreased chloroplast length, chloroplast area, assimilation rate, and stomatal conductance [28]. In the same study, elevated concentrations of leaf volatile organic compounds (VOCs) and other terpenoids were also detected in exposed plants. The same phenomenon was observed in basil (*Ocimum basilicum*) where RF-EMF enhanced VOC emissions and essential oil contents [29].

Notwithstanding the exemplarily mentioned information, several studies have also found that RF-EMF exposure has no effect on plants. An interesting metadata analysis in 2016 revealed that 152 (89.9%) of 169 studies described in the literature from 1995 to 2016 could demonstrate HF-EMF-induced physiological changes, whereas 17 trials (10.1%) showed no change. The results vary greatly depending on exposure frequency, duration, and plant type [10]. It is also worth noting that a significant amount of RF-EMF research is conducted on plants exposed to short durations of radiation and under strictly controlled greenhouse conditions. Only in a few publications, plants were treated with continuous exposure over a longer period or under outdoor conditions [2]. This reflects a significant difference to real-life situations where it is not uncommon for plants to be continuously exposed to RF-EMF for extended periods of time, in some cases throughout their lifetime. Plants are well known for their plasticity—their ability to modify their own physiology in response and adapt to environmental changes. Therefore, to draw correct conclusions regarding RF-EMF effects on plants in the actual world, experiments should be performed under conditions that are as near to those of real world as possible.

In this context, our study intends to examine the effects of RF-EMF exposure on photosynthesis, flowering time, and the expression of two stress-related genes: violaxanthin de-epoxidase (VDE) and zeaxanthin epoxidase (ZEP) in indoor and outdoor environments. Our findings will help to shed light on the potential effects of RF-EMF in plants and their implications in the real world.

## 2. Results

### 2.1. Prolonged RF-EMF Exposure Caused Changes in Plant Fast Chlorophyll Fluorescence Kinetics

We examined photosynthetic efficiency under prolonged RF-EMF exposure using analysis of fast chlorophyll fluorescence kinetics (OJIP analysis). With ten plants in each group, seven separate outdoor trials (four with cultivar *Larissa*, three with cultivar *Briweri*) and one indoor experiment with *Larissa* cultivar were performed from September 2021 to September 2022. Lettuce plants were grown in a radiation-free environment until they were three weeks old, then subjected to continuous RF-EMF exposure. OJIP curves were obtained at the start of treatment (null measurement) and at regular intervals until all plants reached senescence. There were a total of 48 measurement time points. At each measurement time point, at least 40 (field trials) or 100 (greenhouse experiment) measurements were performed, which were evenly distributed between two groups. Analysis of the OJIP curves yielded 50 different photosynthetic-related parameters, which were subsequently used to compare treated and control groups at each measurement time point.

We examined how frequently statistically significant differences in these 50 different OJIP parameters were identified. As depicted in Figure 1, there were few discrepancies between the two groups in the null measurements. In contrast, once the plants were subjected to RF-EMF exposure, significant differences to control groups were observed. The trials with cultivar *Briweri* yielded generally more differences than those with *Larissa*.

### 2.2. Continuous RF-EMF Exposure Resulted in Significant Decrease of Photosynthetic Efficiency in Six out of Seven Outdoor Trials but Not in the Greenhouse Experiment

The frequency analysis in Section 2.1 indicates the differences but it does not allow a comparison of photosynthetic efficiency between the treated and control groups. To answer this question, we employed the integrated biomarker response (IBR) methodology to calculate the photochemical stress index (PSI) [30,31,32]. The PSI is an integrated indicator that reflects the overall photosynthetic efficiency and is computed from 19 separate OJIP variables that correspond to different biochemical processes of photosynthesis [32]. From each measurement, a PSI value is computed. These PSI values were then used to compare the treated and control plants and to determine if the differences were statistically significant. For each time point, a single treated or control PSI ratio was calculated as
(1)$$Treated/control\;PSI-ratio=\frac{Mean\;PSI\;value\;from\;all\;measurements\;of\;treated\;plants\;\;}{Mean\;PSI\;value\;from\;all\;measurements\;of\;control\;plants}$$

The development of treated/control PSI ratios through all trials is displayed in Figure 2. There are no statistically significant differences between treated and control groups in any null measurements, indicating that their photosynthetic efficiency was the same at the start of the trials. No statistically significant differences were detected in the outdoor trial 2 (*Larissa*) and in the greenhouse experiment (*Larissa*) either. In other trials, significant reduction of photosynthetic efficiency was observed in treated plants, particularly towards the end of the experiments. Generally, *Briweri* displayed significant photosynthetic reduction sooner (after two weeks) than *Larissa* (4 weeks). Only in outdoor trial 4 (Larissa) such a reduction was detected in week 2, but it is worth noting that the duration of this trial and of outdoor trial 3 (*Briweri*), both conducted in August 2022, were much shorter than the others, as plants reached senescence after only 2 weeks, whereas the other experiments lasted 4–5 weeks.

We also calculated the Spearman’s rho rank correlation coefficients between the treated/control PSI ratios and exposure duration. In the outdoor trial 2 (*Larissa*) as well as in the greenhouse experiment (*Larissa*), we observed just weak correlations between these two variables. In the other six out of seven field trials, large Spearman’s correlation coefficients (−1.0 < ρ < −0.4) indicate moderate to strong negative correlation between treated/control PSI ratios and the exposure period, implying that in these trials, the photosynthetic performance of the treated plants became progressively worse compared to that of the control groups as the irradiation continued.

### 2.3. Analysis of Individual OJIP Parameters Reveals That Many Photosynthetic Processes Are Simultaneously Affected by RF-EMF

We looked more closely into individual OJIP parameter to find which biochemical process of photosynthesis was affected by RF-EMF exposure. Eight parameters were chosen for analysis: F_V_/F_M_—the maximal quantum yield of PSII photochemistry; F_V_/F_O_—the maximal quantum yield of oxygen-evolving complex (OEC); ΨE_O_—quantum yield of the electron transport in the intersystem electron chain (from Q_A_ to plastocyanin PC); δR_O_—quantum yield of the reduction of end acceptors at PSI side; dVG/dt_O_—excitation energy transfer between the reaction centers; RC/ABS—effective antenna size; RC/CS_O_—reaction center density; area—pool size of reduced plastoquinone (PG) on the reducing side of PS II [33,34,35]. For these parameter, treated/control ratios were also calculated using formulas similar to the one in Section 2.2. A treated/control ratio less than one is usually indicative for lower photosynthetic efficiency than the controls, except for dVG/dt_O_ where the opposite is true: a ratio less than one indicates improved photosynthetic performance [36]. We also performed the Mann–Whitney U test to see if the differences were statistically significant.

Figure 3 depicts the distribution of all the treated/control ratios across all trials and their statistical significance. The detailed time courses for each parameter in each trial are presented in the Appendix A. There was no effect of RF-EMF on ΨE_O_. On the other hand, RF-EMF exposure clearly resulted in a decrease in F_V_/F_M_, F_V_/F_O_, δR_O_,RC/ABS, RC/CS_O_ and an increase in dVG/dt_O_, all indicating a deterioration in photosynthesis compared to the control plants. The effect’s magnitude appears to be much greater in field trials than in greenhouse experiments—only δR_O_ declined to the same extent as in the outdoor trials, whereas other variables decreased only marginally.

We also analyzed profiles of the eight previously described OJIP parameters obtained at the first (null) and the last measurement time points (Figure 4—we substituted dVG/dt_O_ by its reciprocal value 1/dVG/dt_O_ to correlate all parameters positively to photosynthetic efficiency). In cases of outdoor trials 1, 3, 4 (*Larissa*) and outdoor trials 2, 3 (*Briweri*), the simultaneous decrease in many parameters causes the plot area of the last measurements to visibly shrink compared to those of the null measurements. Such reduction indicates an RF-EMF impact on many OJIP parameters related to diverse processes in photosynthesis at the same time. It suggests that the effects of RF-EMF on plant photosynthesis are systemic rather than localized.

### 2.4. Long-Term RF-EMF Exposure Led to Accelerated Flowering Time in Outdoor Environments Only

To study the impact of continuous RF-EMF exposure on flowering time under both indoor and outdoor conditions, we recorded the flowering times of treated and control plants in outdoor trial 3 (*Larissa*—June–July 2022), outdoor trial 4 (*Larissa*—August 2022), and in greenhouse (*Larissa*—September 2022). In addition, we also conducted another flowering time experiment—termed outdoor trial 5—with cultivar *Larissa* under the same conditions in September–October 2022. Figure 5 depicts the flowering time distribution across these four experiments. While in indoor studies there was no difference between two groups, in all three outdoor experiments, a small subset (1–4/10 plants) of the treated plants began flowering considerably sooner than the remainder of the group and plants in the control group. When the flowering time is Z-normalized, the disparity becomes even more pronounced.

### 2.5. RF-EMF Exposure Causes Downregulation of Stress-Related Genes VDE and ZEP

We investigated the effects of RF-EMF exposure on the expression levels of two stress-related genes, i.e., VDE and ZEP. 60 3-week-old lettuce plants (cultivar *Larissa*) were divided into the treated and control group; the former was exposed to RF-EMF. Samples were obtained 6, 12, 24, and 48 h after treatment (three independent replicates). Expression levels of VDE and ZEP were determined by quantitative real-time PCR using actin gene as a reference. Results of gene expression analysis are given in Figure 6. There are considerable variations in the expression patterns of these two genes between the control and RF-EFM-exposed plants. VDE levels in the control plants were similar to the zero level after 6 h (*p* > 0.05), but consistently and statistically significantly higher after 12, 24, and 48 h (*p* < 0.01). In contrast, exposed plants showed an erratic pattern: VDE levels were comparable to zero after 6 h (*p* > 0.05), fell below zero after 12 h, then surged beyond zero after 24 h (*p* < 0.05 in both cases), before returning to zero after 48 h (*p* > 0.05). Similarly, ZEP levels after 6, 12, 24, and 48 h in control plants were statistically significantly greater than zero level (*p* < 0.01). ZEP levels in RF-EMF-exposed plants were higher than zero level after 6 h (*p* < 0.05), equal to zero level after 12 h (*p* > 0.05), higher than zero level again after 24 h (*p* < 0.01), and equal to zero level again after 48 h (*p* > 0.05). As a result of this difference in expression pattern, VDE and ZEP levels in RF-EMF-exposed plants were almost always lower than in the corresponding control plants after 12 h (with the exception of the VDE level after 24 h, when there was no statistically significant difference between the two groups).

### 2.6. RF-EMF Exposure Reduces Photosystem II’s Maximal Photochemical Quantum Yield and Non-Photochemical Quenching under Light Stress Conditions

We postulated that because VDE and ZEP are essential components of the xanthophyll cycle, reduced expression levels of these two genes in RF-EMF-exposed plants would impair plant responses to light stress. To put this hypothesis to the test, we subjected plants to light stress (250 mol/m^2^·s 24/24 h) for 24 h with and without RF-EMF exposure (6 plants/group). Afterward, we investigated the maximum photochemical quantum yield F_V_/F_M_ of Photosystem II and the non-photochemical quenching (NPQ) with Imaging PAM analysis (10 measurements per plant). The findings are shown in Figure 7. Light stress clearly reduced F_V_/F_M_ in both groups, although the decline was substantially greater in the RF-EMF-exposed plants than in the control plants (*p* < 0.01), showing that the latter could withstand light stress better. Furthermore, NPQ induction kinetics analysis revealed that after one day of light stress treatment, NPQ was consistently lower in the exposed plants than in the control groups. A statistically significant difference in NPQ between the two groups could be observed as early as 40 s following the start of illumination (*p* < 0.01).

## 3. Discussion

Our results of field trials suggest that prolonged RF-EMF exposure has a negative impact on plant photosynthesis and development. In six out of seven field trials, HF-EMF exposure induced the progressive reduction of photosynthetic efficiency. Photosynthesis was reduced in both investigated varieties, indicating that this phenomenon is unlikely to be limited to a particular plant cultivar. Individual OJIP parameter analysis revealed a simultaneous decrease in many photosynthetic processes in RF-EMF-exposed plants, particularly towards the end of their life cycle. Such systemic photosynthetic deterioration is usually connected to stress, which typically impacts electron and energy fluxes in and around photosystems PSI and PSII [37]. Another common plant reaction to stress is the early change from vegetative to reproductive development, the formation of flower buds [38]. In our field trials, accelerated flowering was also observed in a subgroup of exposed plants. Together, it can be concluded that exposed plants perceived RF-EMF as a stress factor and reacted accordingly. Thus, on the one hand, our findings are consistent with a growing body of research that confirms the generally harmful effects of RF-EMF on plants. On the other hand, our results from the greenhouse experiments seem to contradict. During the greenhouse experiment, only minor photosynthetic degradation was identified. Just δR_O_—the quantum yield of the reduction of end acceptors at PSI side—was negatively affected by RF-EMF exposure, while other OJIP parameters did not differ. The total photosynthetic efficiency of treated and control plants was comparable, as measured by the integrated PSI parameter. Under greenhouse conditions, there was also no change in flowering time detectable. Based on greenhouse data only, RF-EMF appears to have no discernible effect on lettuce plants under greenhouse conditions.

What can explain this puzzling contradiction? The differences between the locations of the experiments could explain the discrepancies. We postulated that RF-EMF disrupts plant stress response pathways and causes plants to respond insufficiently to common environmental stress (e.g., light, winds, insects, pathogens, rain). The greenhouse environment, where temperature, humidity, and light levels are tightly regulated and the plants are normally sheltered from external influences, might be considered stress free. The plants in the outdoor trials, on the other hand, were continually subjected to widely fluctuating conditions and exposed to a variety of environmental stressors. That explains why stress symptoms (deteriorated photosynthesis, accelerated flowering time) occurred only under the field conditions.

This hypothesis is further corroborated by our results that the two stress-related genes VDE and ZEP were down-regulated upon RF-EMF exposure. In plants, these two genes play important roles in abiotic stress responses. They are crucial components of the xanthophyll cycle which protects the plants from photo-oxidative stress caused by excessively absorbed light energy. Furthermore, VDE and ZEP are key enzymes in the biosynthetic pathway of abscisic acid (ABA), the central plant stress hormone. Overexpression and down-regulation of VDEs and ZEPs have been found in plants to regulate abiotic stress tolerance respectively positive or negative. The *Arabidopsis* VDE-deficient mutant *npq1* is more sensitive against high light stress and cold stress than wild type [39]. The overexpression of various VDE homologs in *Arabidopsis* led to the improvement of stress tolerance against high light stress [40], drought stress [41], and salt stress [42]. Similarly, the *Arabidopsis* ZEP-deficient mutant *aba2* is highly sensitive against water loss through stomata and displays a severe withering phenotype [43], while the overexpression of ZEP in *Arabidopsis* enhances resistance against high light stress, cold stress [44], and osmotic stress [45]. In our experiment, lettuce plants exposed to RF-EMF had lower levels of VDE and ZEP, which likely made them more susceptible to stress.

Because the RF-EMF-exposed plants had lower levels of VDE and ZEP, it was hypothesized that they were more susceptible to light stress than the control plants. Our data supported this hypothesis. F_v_/F_M_, which is generally indicative of plant health status, was much lower in the exposed group than in the control group after one day of light stress treatment. It was also revealed that plants exposed to RF-EMF had consistently lower NPQ levels and the differences could be observed very early. NPQ is the overall term for many plant mechanisms that dissipate excessively absorbed light energy. The processes include the energy-dependent quenching (qE), state transitions (qT), and reversible Photosystem II inactivation (qI). qE, the fastest component, is related to xanthophyll cycle [46]. Our NPQ results are thus also consistent with lower VDE and ZEP levels. The susceptibility of RF-EMF-exposed plants to light stress may also explain the disparities in the data from the outdoor trials. Plants in open areas usually experience light stress at noon (midday depression). Meteorological data (Appendix A) from outdoor trial 2 (*Larissa*) demonstrated normal late autumn weather with numerous overcast and wet days and generally low light intensity. Accordingly, even after 4 weeks of culture, no photosynthetic degradation was seen. On the other hand, the light intensity was very high in outdoor trial 4 (*Larissa*), and photosynthesis reduction was observed as early as week 2.

In our gene expression analysis, the levels of VDE and ZEP in the control group increased steadily, most likely as the result of continuous illumination, while in RF-EMF-exposed plants, their expression patterns were rather erratic. It is still unclear what caused this behavior. Such pattern is somewhat similar to the cyclic expression behavior of many plant transcription factors in response to abiotic stress [47,48]. According to the network pulse model, this pattern is conditioned by changing RNase activity [49] and represents a coordinated strategy against stress, in which stress is initially countered with early, rapid reactions, followed by slower, more comprehensive, and more appropriate responses [48]. Roux et al. (2007) also observed a rapid, transient accumulation of the stress-related genes calm-n6 and pin2 within 15 min of exposure to 900 MHz RF-EMF, followed by a decline to basal levels after 30 min and a second increase after 60 min [12]. The obvious difference in kinetics, on the other hand, suggests that the network pulse hypothesis is unlikely to be the direct cause of the aforementioned expression pattern. In the future, more research would be necessary to elucidate the underlying regulatory elements of this phenomenon.

Our findings highlight the need for scientific studies on the effects of RF-EMF in plants under realistic scenarios. Significant RF-EMF effects could only be detected under outdoor settings. If we had conducted our experiments exclusively in the greenhouse, these effects would have gone unnoticed. Furthermore, the potential interaction between RF-EMF and other stressors requires further, more thorough investigation. Such connections, if verified, would have major consequences. We could expect in the near future both, the massively increased applications of wireless devices as well as a significant upsurge in weather extremes and insect outbreaks because of climate change. These two factors and their interconnections might have significant impacts for plants and endanger both our food security and the stability of our ecosystems.

## 4. Material and Methods

### 4.1. Overview of the Experimental Plan

The experiments were conducted between September 2021 and September 2022. Overall, we investigated the expression of two stress genes VDE and ZEP, flowering time, and kinetics of fast chlorophyll fluorescence under the influence of RF-EMF compared to control plants. The details of these experiments are shown in Table 2 and in the following sections.

### 4.2. Plant Cultivation

*Lactuca sativa* (cultivars *Larissa* and *Briweri*) was cultivated in soil pots under greenhouse conditions. Temperature was maintained between 19 °C–23 °C with relative humidity at 50–60%. RF-EMF strength in the greenhouse was minimal. Three weeks after sowing, young plants were transferred to a phytochamber (indoor experiments—temperature 20 °C, relative humidity 50%, light intensity 100 µmol/m^2^·s, light/dark 16/8 h) or to the experimental field of Forschungsring e.V. (indoor experiments—coordinates 49°50′10″ N 8°4′25″ E). With the exception of the gene expression analysis, where plants were harvested immediately after exposure, plants continued to be cultivated until senescence in all other experiments.

### 4.3. RF-EMF Exposure

To investigate the RF-EMF effects on plants, we generated electromagnetic fields with frequency ranges of 1880–1900 MHz (DECT) and 2.4 and 5 GHz (Wi-Fi). RF-EMF was induced by two Wi-Fi systems (Fritzbox 7530) with a DECT base station and two DECT phones (Motorola t412+, with eco-mode switched off). The DECT phones were in permanent telephone connection with an additional terminal. The RF-EMF radiation was measured with two broadband RF analyzers (Gigahertz Solution). The RF-EMF radiation (peak values) in the frequency range 1880–1900 MHz and 2.4 GHz measured with the high frequency analyzer (HF59B for frequencies from 700 Mhz–2.7 GHz) was 8000 μW/m^2^. The upper Wi-Fi band (5 GHz) measured with the high frequency analyzer (HFW35C, Gigahertz Solutions, Germany for Frequencies 2.4–6 GHz) was 2000 μW/m^2^. The power flux densities in our experiments are comparable to the usual level in a city center [50].

The RF-EMF exposure was conducted on the Forschungsring e.V. experimental field (coordinates 49°49′57.4″ N 8°34′22.2″ E) or in a phytochamber in the Technical University of Darmstadt’s greenhouse. The *Lactuca sativa* plants were separated into two groups of 9–10 plants each at the start of the experiment. Fast chlorophyll fluorescence kinetics were measured and statistically analyzed after 48 h of acclimatization to ensure that the experimental groups were comparable. The RF-EMF emitters were then turned on, and the plants were measured at regular intervals (about twice—three times a week) until senescence.

The 10 control plants (Figure 8) placed adjacent to the RF-EMF-exposed area were shielded from the adjoining RF-EMF emitters with a fine-mesh metal fence (mesh size 13 mm and height 120 cm) and therefore served as a reference in the outdoor experiments. The RF-EMF-exposed plants were approximately 8 m apart from the control plants. In the greenhouse studies, the control groups were housed in another separate phytochamber. In both cases, the RF- and high frequency analyzers were used to validate the minimal level of RF-EMF that the control groups were exposed to.

### 4.4. Gene Expression Analysis

Sixty three-week-old *Lactuca sativa* plants were separated into two groups of 30 plants each and placed in two identical phytochambers (temperature 20° degrees Celsius, relative humidity 50%, light intensity 100 µmol/m^2^·s 24/24 h). After 24 h of acclimating to the new sites, RF-EMF was activated in one phytochamber while conditions in the other remained unchanged. Six plants were collected from both treated and control plants immediately before and 6 h, 12 h, 24 h, and 24 h after RF-EMF activation. These six plants were separated into three samples of two plants each and immediately frozen in liquid nitrogen (Figure 9). From each sample, total RNA was extracted and used for cDNA synthesis, which was subsequently used for quantitative real-time PCR (qPCR).

The frozen plant material was crushed in a mortar and pestle in the presence of liquid nitrogen. Approximately 100 mg was transferred to a pre-cooled Eppendorf tube and mixed well with 800 µL of lysis buffer (0.6 M NaCl, Tris/HCl, 0.01 M EDTA, 0.1% SDS, pH 8.0, treated with 0.1% DEPC). The mixture was extracted with the same volume of phenol-chloroform-isoamyl alcohol (25/24/1 *v/v/v*) for one hour. In the next step, the tubes were centrifuged and the clear aqueous phase was collected in a new Eppendorf tube, mixed with 600 µL LiCl 8 M, and incubated overnight at 4 °C. The next day, the RNA precipitate was collected by centrifugation and treated with DNase I at 37 °C for 30 min before being purified again by ethanol precipitation. The RNA was finally dissolved in DEPC-treated water.

An amount of 1 µg of total RNA was used to synthesize cDNA using MuLV reverse transcriptase from New Englands Biolabs. cDNA was diluted 1:10 and then used for PCR. qPCR was performed using the StepOne Plus Real-time PCR System and 2× qPCRBIO SyGReen Mix Hi-ROX (PCRBIO). All reactions were performed according to the manufacturer’s instructions.

In qPCR reactions, primers for the reference gene actin as well as for two stress genes VDE and ZEP were used. The primer sequences are given in Table 3. The 2^−∆∆Ct^ method was applied to calculate and compare the relative transcript amounts of VDE and ZEP between samples.

### 4.5. Flowering Time Analysis

The time of flowering was determined as the time when flower became visible (i.e., the opening of the bracts). Flowering times were counted from the start of the RF-EMF treatment. Z-normalization of the flowering time was performed with the following formula:(2)$$Z-\mathrm{normalized}\;\mathrm{flowering}\;\mathrm{time}=\;\frac{\mathrm{Flowering}\;\mathrm{time}-\mathrm{Average}\;\mathrm{flowering}\;\mathrm{time}\;\left(\mathrm{control}\;\mathrm{group}\right)}{\mathrm{Standard}\;\mathrm{deviation}\;\left(\mathrm{control}\;\mathrm{group}\right)}$$

### 4.6. Measurements of Fast Chlorophyll Fluorescence Kinetics

The Pocket PEA device (Hansatech) was used to measure the leaf’s fast chlorophyll fluorescence kinetics. Measure time was one second, and the saturating light intensity was 3500 µmol/m^2^·s. Under outdoor conditions, measurements were performed at least 3 h after sunset. In the greenhouse, measurements were obtained at the end of the dark phase. OJIP parameters were calculated using PEA Plus software Version 1.13 (Hansatech) with F1-Mode. The photochemical stress index (PSI) was calculated from 19 separate OJIP variables as described in other publications [30,31,32] and in the Appendix A.

### 4.7. Light Stress Experiment and Imaging PAM Analysis

As with prior experiments, two greenhouse phytochambers were set up for the light stress experiment. RF-EMF was turned on in one phytochamber (exposed group), but not in the other (control group). Light stress was induced by maintaining a high light intensity of 250 mol/m^2^·s continuously for 24 h. Three-week-old plants were transferred to two phytochambers and treated for 24 h. The experiment was conducted twice with three plants per group and repetition. Photosystem II‘s maximum photochemical quantum yield F_V_/F_M_ and non-photochemical quenching (NPQ) were measured using quenching analysis with Imaging PAM MAXI before and after light stress treatment (Walz). NPQ was induced by blue light illumination at 1250 mol/m^2^·s for 10 min after plants were dark-adapted for at least one hour. Saturation pulses were given every 20 s. PAM data were processed using ImagingWin Software Version 2.56 (Walz) on 10 randomly selected spots per plant.

### 4.8. Statistical Analysis

Due to the lack of data normality, the non-parametric Mann–Whitney U test was applied to compare treated and control groups throughout all experiments. Statistical significance was assumed when *p* < 0.05. Statistical tests were performed with Jamovi (Version 1.6.23) and Microsoft Excel (Microsoft Office Professional Plus 2013) with Real Statistics Add-ins.

## Figures and Tables

**Figure 1 plants-12-01082-f001:**
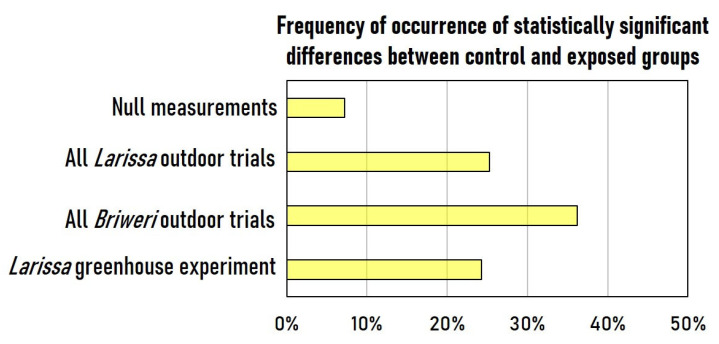
The degree of variation between the control and RF-EMF-exposed groups in OJIP analysis. The graph represents the frequency with which a statistically significant difference could be found when comparing all 50 OJIP variables of the control and exposed plants measured on the same day. Null measurements are those obtained before the RF-EMF is activated. Changes in both directions are counted regardless of whether the value increased or decreased. While the difference between two groups was minimal in the null measurements, the degree of variance rose once the RF-EMF was turned on.

**Figure 2 plants-12-01082-f002:**
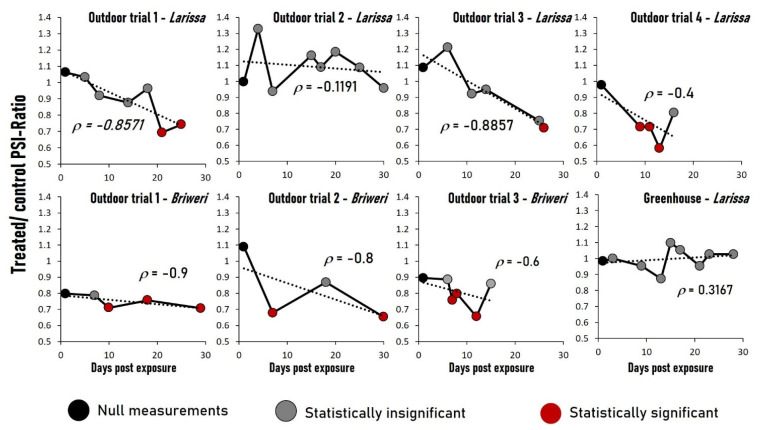
The courses of treated/control PSI ratios across the measurement time points in all trials. There are no statistically significant differences between treated and control groups in any null measurements. Black dots: null measurement. Gray dots: measurements with no statistically significant differences between treated and control groups. Red dots: measurements with statistically significant differences between treated and control groups. Also shown are the Spearman’s rho rank correlation coefficients ρ between the exposure time and the treated/control PSI ratios. ρ interpretation: −0.2 < ρ < 0.2: no correlation; −0.4 < ρ < −0.2 and 0.2 < ρ < 0.4: weak correlation; −0.6 < ρ < −0.4 and 0.4 < ρ < 0.6: moderate correlation; −1.0 < ρ < −0.6 and 0.6 < ρ < 1.0: strong correlation.

**Figure 3 plants-12-01082-f003:**
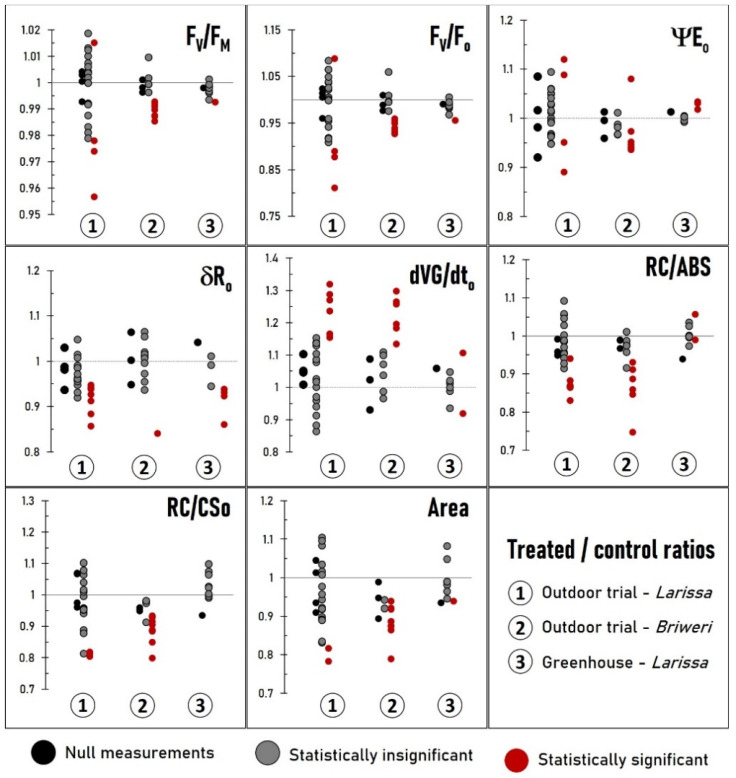
Distribution of treated/control ratios concerning the eight OJIP parameter under investigation: F_V_/F_M_—the maximal quantum yield of PSII photochemistry; F_V_/F_O_—the maximal quantum yield of oxygen-evolving complex (OEC); ΨE_O_—quantum yield of the electron transport in the intersystem electron chain (from Q_A_ to plastocyanin PC); δR_O_—quantum yield of the reduction in end acceptors at PSI side; dVG/dt_O_—excitation energy transfer between the reaction centers; RC/ABS—effective antenna size; RC/CS_O_—reaction center density; area—pool size of reduced plastoquinone (PG) on the reducing side of PS II. The results are separated into three groups corresponding to outdoor trials with cultivar *Larissa* (1), outdoor trials with *Briweri* (2) and greenhouse experiment with *Larissa* (3). A dot symbolizes the treated/control ratio of each measurement point, in other words the difference between two groups at each measurement day: black—null measurements; gray—no statistically significant difference between treated and control groups (*p* > 0.05); red—statistically significant difference from two groups (*p* < 0.05). A treated/control ratio larger than one in case of dVG/dt_O_ or less than one in case of all other parameters indicates lower photosynthetic efficiency in treated plants compared to control plants.

**Figure 4 plants-12-01082-f004:**
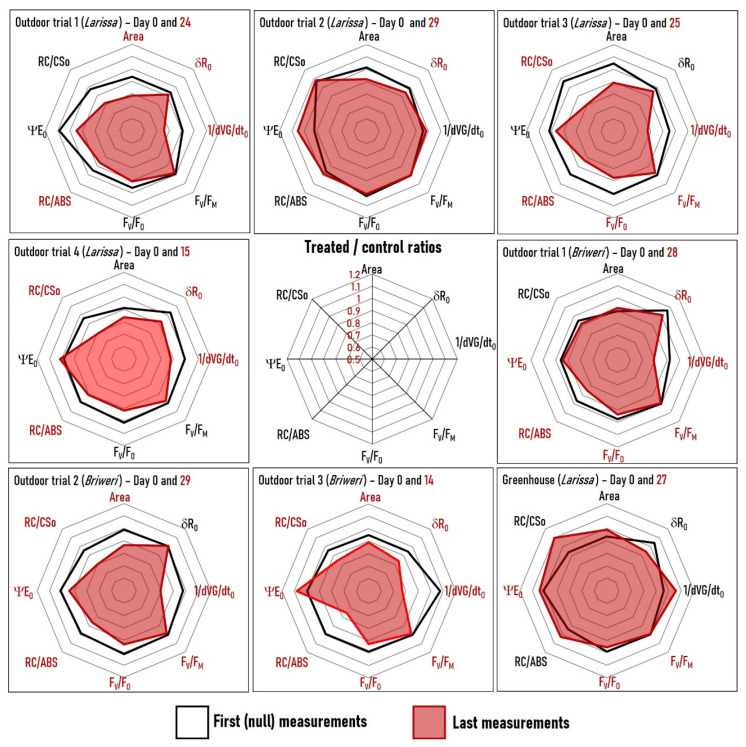
Spider plots depicting the treated/controls ratio at the first (null) and the last measurement time points for the eight OJIP parameters stated earlier. We substituted dVG/dt_O_ with its reciprocal value 1/dVG/dt_O_ so that all parameters correlate positively with photosynthetic efficiency. The plot’s scale is displayed in the center. If a parameter’s label is red, the last measurement revealed a statistically significant difference (*p* < 0.05) between RF-EMF-exposed and control plants with relation to that particular parameter.

**Figure 5 plants-12-01082-f005:**
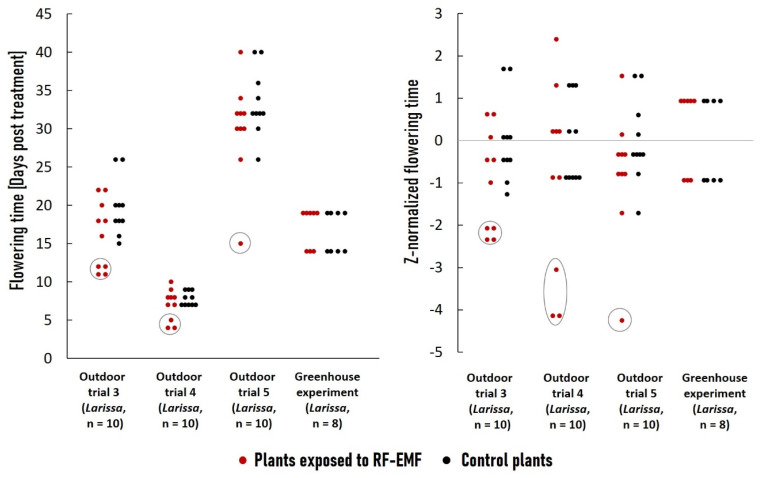
Flowering time comparison of plants continuously exposed to RF-EMF and controls. In total, there were 3 independent experiments under outdoor conditions (outdoor 1, 2, 3) and one under indoor conditions (greenhouse), each with equal numbers (8–10) plants from each group. The dots represent flowering time of individual plants. (**Left**) Flowering time (days after experiment onset). (**Right**) Normalized flowering time based on the mean and standard deviation of the control group’s data (Z-scores). The subsets of treated plants that began flowering substantially earlier than the rest of the group and plants in the control group are indicated by circles.

**Figure 6 plants-12-01082-f006:**
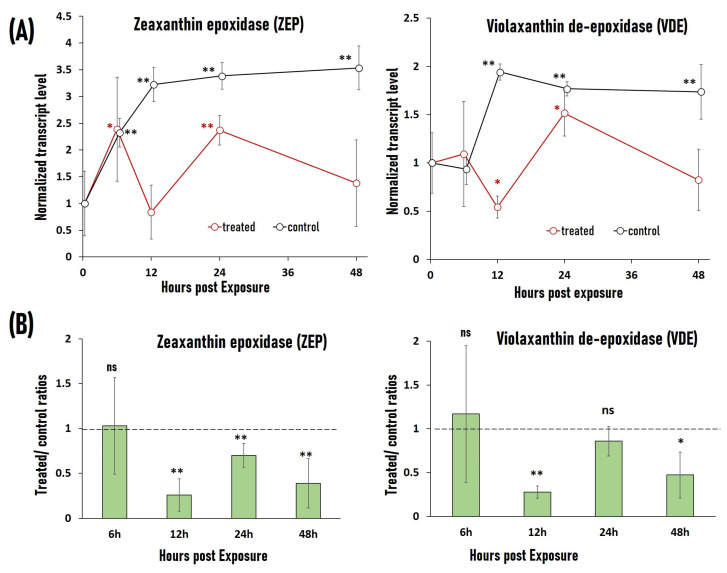
Gene expression analysis of stress genes violaxanthin de-epoxidase (VDE) and zeaxanthin epoxidase (ZEP) in lettuce plants exposed to RF-EMF. The reference gene is actin. (**A**) Normalized transcript levels of VDE and ZEP at 0, 6, 12, 24, and 48 h after RF-EMF treatment. The average transcript levels of all plants at Time 0 (start of the RF-EMF treatment) are determined as 1. Star symbols (*—0.01 < *p* < 0.05; **—*p* < 0.01) denote the statistically significant differences to VDE/ZEP levels at t = 0 (threshold *p* = 0.05). Error bars represent 95% confidence interval (n = 3). (**B**) The treated/control ratio of VDE and ZEP transcript levels at 6, 12, 24, and 48 h after RF-EMF treatment. Statistical symbols: ns—no statistically significant differences between control and exposed groups (*p* > 0.05), *—statistically significant differences (0.01 < *p* < 0.05), **—statistically highly significant differences (*p* < 0.01). Error bars represent 95% confidence interval (n = 3). Dotted line marks the position of treated/control ratio equal 1.

**Figure 7 plants-12-01082-f007:**
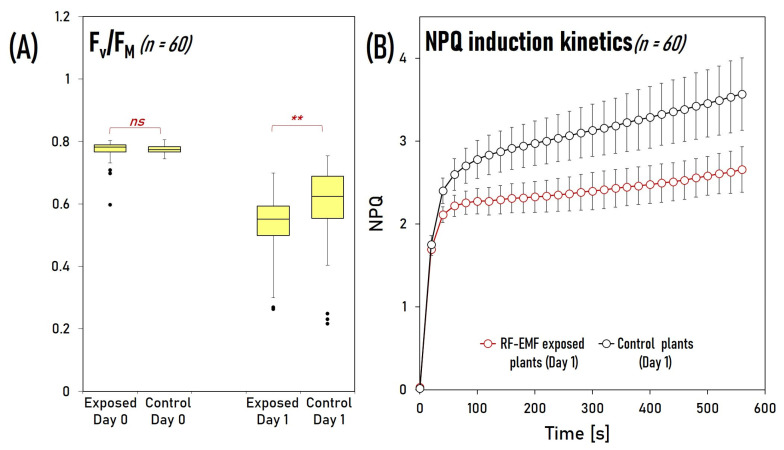
Comparison of the maximum photochemical quantum yield F_V_/F_M_ of Photosystem II and the non-photochemical quenching (NPQ) of RE-EMF-exposed and control plants after one day of light stress treatment. (**A**) F_V_/F_M_ of exposed and control plants before (Day 0) and after treatment (Day 1). Statistical symbols: ns—no statistically significant differences between control and exposed groups (*p* > 0.05), **—statistically highly significant differences (*p* < 0.01). (**B**) NPQ induction kinetics of exposed and control plants after one day light stress treatment. Error bars represent 95% confidence interval (n = 60). In all cases where the error bars do not overlap, the difference is statistically highly significant (*p* < 0.01).

**Figure 8 plants-12-01082-f008:**
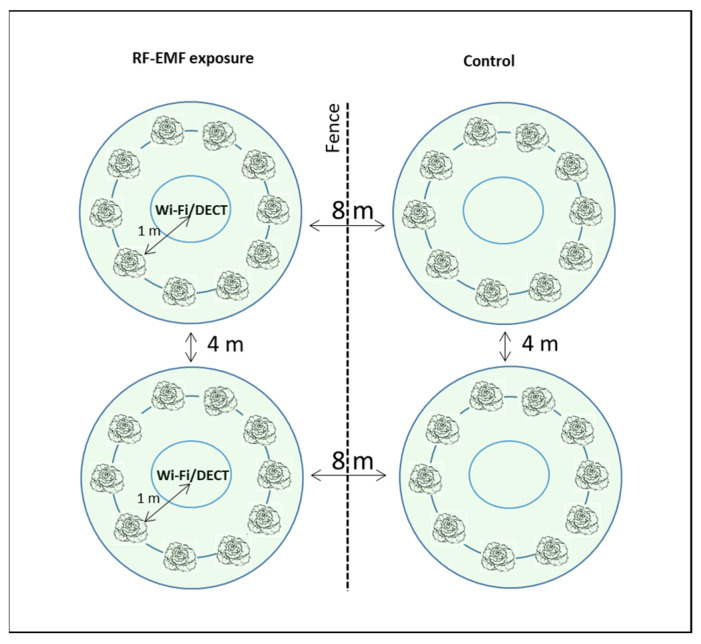
Experimental setup—RF-EMF exposure of lettuce plants in the outdoor experiments.

**Figure 9 plants-12-01082-f009:**
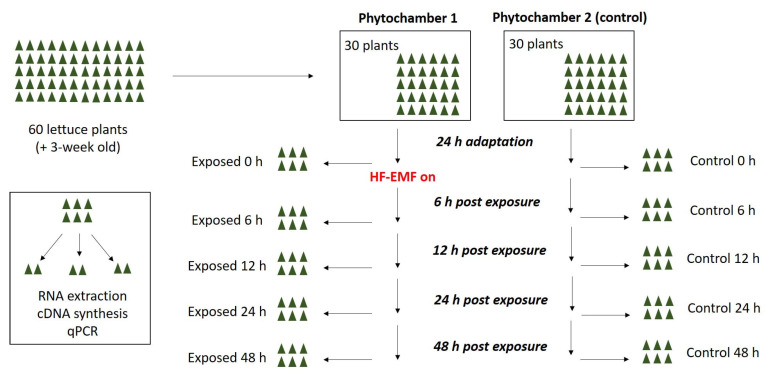
Experimental plan—expression analysis of stress genes ZEP and VDE under RF-EMF exposure.

**Table 1 plants-12-01082-t001:** Frequency ranges radiated by natural and manmade RF-EMF sources [2].

RF-EFM Sources	Frequency Range
Natural	
Lightning discharge	<30 MHz
Sun	>30 MHz
Man-made	
AM radio	550–1600 MHz
FM radio	88–108 MHz
Mobile phones	800–5500 MHz
Wi-Fi system	2.4 GHz, 5.8 GHz
Smart meters	902–928 MHz
Microwave ovens	2.45 GHz

**Table 2 plants-12-01082-t002:** Experiments in the study with their time, location, used cultivars, number of plants (treated and control), RF-EMF exposure duration and analysis (G: gene expression analysis; F: flowering time analysis; O: analysis of fast chlorophyll fluorescence OJIP kinetics; P: Imaging PAM analysis).

Name	Time	Location	Lettuce Cultivar	Number of Plants (Treated/Control)	Exposure Duration	Analysis
Gene expression analysis	August 2022	Indoor	*Larissa*	30/30	Up to 48 h	G
Outdoor trial 1 (*Larissa*)	August–September 2021	Outdoor	*Larissa*	10/10	4 weeks	O
Outdoor trial 2 (*Larissa*)	October–November 2021	Outdoor	*Larissa*	10/10	5 weeks	O
Outdoor trial 3 (*Larissa*)	June–July 2022	Outdoor	*Larissa*	10/10	4 weeks	F, O
Outdoor trial 4 (*Larissa*)	August 2022	Outdoor	*Larissa*	10/10	2 weeks	F, O
Outdoor trial 5 (*Larissa*)	September–October 2022	Outdoor	*Larissa*	10/10	6 weeks	F
Outdoor trial 1 (*Briweri*)	June–July 2022	Outdoor	*Briweri*	10/10	4 weeks	O
Outdoor trial 2 (*Briweri*)	June–July 2022	Outdoor	*Briweri*	10/10	4 weeks	O
Outdoor trial 3 (*Briweri*)	August 2022	Outdoor	*Briweri*	10/10	2 weeks	O
Greenhouse experiment (*Larissa*)	September 2022	Indoor	*Larissa*	9/9	4 weeks	F, O
Light stress experiment	February 2023	Indoor	*Larissa*	6/6	24 h	P

**Table 3 plants-12-01082-t003:** Primers for gene expression analysis.

Primer	Sequence
Actin forward primer	TAC ATG TTC ACC ACC ACA GC
Actin reverse primer	ATG AGC TGG ACT TGG CGG TTT
VDE forward primer	CGT TTC GAC CTC GGT AGT ATA CG
VDE reverse primer	CTG CTG TGC AAA CAT TTG TTC AAG
ZEP forward primer	TCT GAT GTT GGG GGT GGA AA
ZEP reverse primer	TCC GTC GCA AGC AAC AAA T

## Data Availability

OJIP data are available in the Appendix A.

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
