# Peer review of "Impacts of Radio-Frequency Electromagnetic Field (RF-EMF) on Lettuce (Lactuca sativa)—Evidence for RF-EMF Interference with Plant Stress Responses"

_plants, 2023, doi:10.3390/plants12051082_

Round 1
Reviewer 1 Report
This manuscript experimentally investigated the influence of RF-EMF on plants and implied that RF-EMF might interfere with plant stress responses. In particular, this manuscript highlighted the difference between the greenhouse and outdoor conditions in experimental results which were often missed in previous studies. This manuscript is well written both in structure and language, hence I recommend it to be accepted.
Author Response
Thank you for your kind words. The positive comments are much appreciated.
Reviewer 2 Report
1) The title of the article should be changed by shortening it.
2) The captions in Fig. 1 are too small, they need to be changed.
3) Introduction needs to be expanded. In this text, it is not clear why this work is being carried out, it is not clear whether the magnetic field of such frequencies can be present in field and greenhouse conditions, and in what cases. Additional information is needed about the studied frequency ranges of the electromagnetic field, as well as an indication of the intensities that act in nature, only the regular level in the city center is shown (line 379). Can the magnetic field of the Schumann resonance be superimposed on the one you are studying in the field experiment in the presence of a fine-mesh metal fence. Was fence only on the side or also on top of plants?
4) What is the difference between the 2 varieties presented Larissa and Briwari? Why the Briwari cultivar was not investigated in the greenhouse experiment?
5) It is better to use the same type of designations throughout the text: WLAN (for example, line 370), or Wi-Fi (line 15); ZE (Fig. 8), or ZEP (line 418).
6) It is known that the planting month can affect the flowering of plants, so I think that it is more correct to compare "Greenhouse experiment" only with "Outdoor trial 1". Since there was no difference between the two groups of plants in indoor studies (Fig. 5), this suggests that the change in flowering time is not due to the action of the magnetic field, but to other external factors. The conclusions are very hypothetical and almost not supported by comparisons with scientific articles (lines 299-307). The discussion section needs to be extensively reworked. Explanation contradicts the facts (lines 284-291). Logically, if the magnetic field operates in the presence of other stress factors and its action blocks the response pathways to stress, then plants should not start flowering faster, on the contrary, flowering should occur faster in conditions when there is only a magnetic field, which is also stressful for the plant.
7) If you say that ABA is the most important stress hormone (line 314), then why hasn't the content of this hormone in plants or the expression of the genes responsible for its production been investigated? Is there a correlation between photosynthesis rates and enzymes changes? The proposed mechanisms through which the action of electromagnetic field leads to changes in gene expression of ZEP and VDE described too superficially. The discussion section needs to be extensively reworked.
8) The design of the list of references does not comply with the rules of the journal.
Author Response
Comment 1: The title of the article should be changed by shortening it.
Answer: We changed the title of our article to “Impacts of radio frequency electromagnetic field on lettuce (Lactuca sativa) – Evidence for RF-EMF interference with plant stress responses”
Comment 2: The captions in Fig. 1 are too small, they need to be changed.
Answer: Another reviewer also commented on the brevity of Figure 1's caption. We have expanded on it in the revised manuscript. Figure 1 has also been simplified for clarity's sake.
Comment 3: Introduction needs to be expanded. In this text, it is not clear why this work is being carried out, it is not clear whether the magnetic field of such frequencies can be present in field and greenhouse conditions, and in what cases. Additional information is needed about the studied frequency ranges of the electromagnetic field, as well as an indication of the intensities that act in nature, only the regular level in the city center is shown (line 379).
Answer: We included a table that summarizes the frequency ranges radiated by several frequently occurring natural and man-made RF-EMF sources.
Comment 4: Can the magnetic field of the Schumann resonance be superimposed on the one you are studying in the field experiment in the presence of a fine-mesh metal fence. Was fence only on the side or also on top of plants?
Answer: Because the metal barrier that separated the control and exposed groups was only on the side, the electromagnetic field of the Schumann resonance could easily be superimposed on the examined RF-EMF settings. However, because the distance between the two groups was very close, the Schumann-EMF on each group is expected to be equal and thus does not affect our experiments.
Comment 5: What is the difference between the 2 varieties presented Larissa and Briwari? Why the Briwari cultivar was not investigated in the greenhouse experiment?
Answer: Our primary research subject is the cultivar Larissa. A comprehensive set of experiments were carried out with this cultivar, including gene expression analysis, flowering time analysis, photosynthesis research and stress study. However, there are many other lettuce cultivars currently being utilized in agriculture, and it is uncertain if the results from Larissa would be equally applicable to other cultivars. As stated in the introduction, we aimed to conduct our experiments as closely as possible to real-life situations. As a result, it was decided that another cultivar, Briweri, should also be investigated in the outdoor trials. Briweri and Larissa are largely distinguished by their appearance and flavour. We are not aware of any study that systematically examined the growth performance and stress tolerance of these two cultivars.
It would have been ideal to investigate the cultivar Briweri under greenhouse conditions as well, however this was unfortunately not feasible due to a lack of staff and phytochamber availability. For example, two available phytochambers in the greenhouse were reserved for the gene expression experiment for most of our research, where the experimenter had to overcome several technical challenges to obtain satisfactory results. It was only in the last few months that we were able to use these phytochambers for the long-term exposure experiment. However, we are confident that the mentioned omission does not affect the validity of our conclusions.
Comment 6: It is better to use the same type of designations throughout the text: WLAN (for example, line 370), or Wi-Fi (line 15); ZE (Fig. 8), or ZEP (line 418).
Answer: We agree. We have made the necessary changes.
Comment 7: It is known that the planting month can affect the flowering of plants, so I think that it is more correct to compare "Greenhouse experiment" only with "Outdoor trial 1".
Answer: You are correct that the time of year has a considerable impact on flowering time, as seen in Figure 5. However, in our study, we always examined and compared the flowering time of control and RF-EMF exposed plants that were grown concurrently and under the same conditions. As a result, they can be compared regardless of the month of planting.
Comment 8: Since there was no difference between the two groups of plants in indoor studies (Fig. 5), this suggests that the change in flowering time is not due to the action of the magnetic field, but to other external factors. The conclusions are very hypothetical and almost not supported by comparisons with scientific articles (lines 299-307). The discussion section needs to be extensively reworked.
Answer: We provided further data that clearly demonstrate the influence of RF-EMF on plant stress responses to high light stress.
Comment 9: Explanation contradicts the facts (lines 284-291). Logically, if the magnetic field operates in the presence of other stress factors and its action blocks the response pathways to stress, then plants should not start flowering faster, on the contrary, flowering should occur faster in conditions when there is only a magnetic field, which is also stressful for the plant.
Answer: The reviewer's statement is based on two assumptions: (1) RF-EMF is stressful for plants, and (2) RF-EMF always blocks stress response pathways. Both of these cannot be taken for granted. Our findings indicated that RF-EMF exposure under the experimental conditions was not stressful for plants. While this finding contradicts numerous previously published studies that found RF-EMF exposure to be stressful for plants, it is hardly surprising. As noted in the introduction, there are significant differences in terms of exposure frequency, duration, and plant species in these studies and in our experiments. Furthermore, the complex relationship of RF-EMF exposure, stress response pathways, and flowering defies simplistic "one-size-fits-all" models. There is evidence that RF-EMF does not interact evenly with different types of stress. While our new data show RF-EMF interference with high light stress response, our other data show that the RF-EMF effect was minimal under cold stress, as seen in Outdoor trial 2. We believe that RF-EMF interaction with plant stress response is a highly complicated mechanism that will require extensive future investigation.
Comment 10: If you say that ABA is the most important stress hormone (line 314), then why hasn't the content of this hormone in plants or the expression of the genes responsible for its production been investigated?
Answer: There are several approaches for studying plant stress. We selected chlorophyll fluorescence rapid kinetic analysis (OJIP analysis) because it is quick, sensitive, and non-invasive. The ABA assay would have necessitated plant destruction, increasing the number of needed plants many-fold.
We will certainly want to check at ABA levels and ABA pathways in our next study when we explore the impact of RF-EMF on stress reactions.
Comment 11: Is there a correlation between photosynthesis rates and enzymes changes? The proposed mechanisms through which the action of electromagnetic field leads to changes in gene expression of ZEP and VDE described too superficially. The discussion section needs to be extensively reworked.
Answer: VDE and ZEP are crucial components of the xanthophyll cycle which protects the plant’s photosynthetic machinery from photo-oxidative stress caused by excessively absorbed light energy. With the new data, we could construct a better, more thorough explanation for how RF-EFM interferes with plant stress response. However, our results do not explain why ZEP and VDE were down-regulated. Answering this question would need substantial future research.
Comment 12: The design of the list of references does not comply with the rules of the journal.
Answer: We made the necessary changes.
Reviewer 3 Report
I definitely agree with the authors, that their „findings highlight the need of scientific studies on effect of RF-EMF in plant under realistic scenarios”.
However, In my opinion the paper is not yet ready for publication.
First - I know, that presentation and interpretation of OJIP generated parameters might be a nightmare. However, could Authors make it easier for the readers? I spend a long time, trying to understand what actually is shown in Figure 1, and after all, I’m not sure, I understood. What is compared with what? What is the zero level for evaluation of change? If this is a null experiment, why there are also some changes? Also - the change seems to be any change, so finally - the figures shows increase in the variation of a feature, not the actual trend in any direction?
Methodology for OJIP - how many measurement for one data point on one plant were done? which leaves were selected?
Line 193-195 - if you cannot conclude about the increase or decrease, it means - there is no change. You may just have and increase in variation.
Figure 2 - the changes seems to correlate with the climatic data shown in supplementary. Maybe it would be reasonable to add some overlay, at least in SI?
Figure 3. again, why data are presented like this? Why to mark “significantly different plants”? In a statistical analysis, usually the group is treated as a population, and you conclude about a difference between population. Therefore, there are no “significantly different” individuals. How the treated/control ratio is calculated, if a control group consist of many plants? What is actually null experiment standing for, if plants were younger than analyzed once?
Why actually only ZEP and VDE genes were chosen? It seems a bit arbitrary. Anyway, conclusions from transcripts level only is not enough. ZEP and VDE activity can be easily assayed. Additionally, if these proteins are really affected, there should be a change in photosynthetic pigments also, at least of the xanthophyll pool.
I cannot agree, with presented data points, with “cyclical change” the authors describe (line 328), and, especially, the parallels built for pulse models of RNAse activity. The mechanisms for cycle occurring in minutes cannot be simply transferred into mechanisms for cycle in days. Anyway, the changes in VDE/ZEP transcript level may correlate with photoperiod (which is not marked, why?).
Figures 6 legends says “the average transcripts level at time 0 are determined as 1”/ Why then the zero level is 0.75 and 1.25?
Author Response
Comment 1: I definitely agree with the authors, that their „findings highlight the need of scientific studies on effect of RF-EMF in plant under realistic scenarios”.
However, in my opinion the paper is not yet ready for publication.
First - I know, that presentation and interpretation of OJIP generated parameters might be a nightmare. However, could Authors make it easier for the readers? I spend a long time, trying to understand what actually is shown in Figure 1, and after all, I’m not sure, I understood. What is compared with what? What is the zero level for evaluation of change? If this is a null experiment, why there are also some changes? Also - the change seems to be any change, so finally - the figures shows increase in the variation of a feature, not the actual trend in any direction?
Answer: Figure 1's purpose is to demonstrate that there were numerous statistically significant variations in various OJIP parameters between the control and RF-EMF exposed plants. There were some discrepancies during null measurements (i.e. before the application of RF-EMF), but the frequency of differences grew significantly in all parameters once the RF-EMF was turned on.
To answer your questions:
“What is compared with what?” => The results of control and RF-EMF exposed plants measured on the same day (i.e. measurement time point)
“What is the zero level for evaluation of change?” => Comparison between control and RF-EMF exposed plants measured at the beginning of the experiment before RF-EMF was activated (i.e. null measurements).
“If this is a null experiment, why there are also some changes?” => Because OJIP analysis is a very sensitive method capable of detecting the smallest differences in photosynthetic performance. Thus there are small degree of variance between two groups even before the activation of RF-EMF. The level of variation increased significantly after RF-EMF was turned on.
“Also - the change seems to be any change, so finally - the figures shows increase in the variation of a feature, not the actual trend in any direction?” => That is correct. Admittedly, it is a very crude frequency analysis that completely disregard the direction of changes – as you correctly pointed out. As we stated in Lines 137-138, indicates for differences but it does not allow a comparison of photosynthetic efficiency.
We decided to replace Figure 1 with a simple bar chart illustrating the degree of variance between two groups at null measurements and subsequently, to prevent unnecessary confusion.
Comment 2: Methodology for OJIP - how many measurement for one data point on one plant were done? Which leaves were selected?
Answer: At each measurement time point (i.e. day of measurement), 40 measurements in field trials (one measurement per leaf, two measurements per plant) or 108 measurements in greenhouse experiment (two measurement per leaf, six measurements per plant) measurements were performed, which were evenly distributed between two groups. Leaves of the same age were labelled at the beginning of the experiment (by then they were 3 week old) and continuously measured. If a single leaf was damaged or developed lesions, all measured leaves in all plants were exchanged by moving to the leaves directly above them.
Comment 3: Line 193-195 - if you cannot conclude about the increase or decrease, it means - there is no change. You may just have and increase in variation.
Answer: We agree.
Comment 4: Figure 2 - the changes seems to correlate with the climatic data shown in supplementary. Maybe it would be reasonable to add some overlay, at least in SI?
Answer: We performed additional experiment with which the photosynthetic deterioration could be linked to light stress (see our Answer 7). We added an overlay in the Supplementary Materials.
Comment 5: Figure 3. again, why data are presented like this? Why to mark “significantly different plants”? In a statistical analysis, usually the group is treated as a population, and you conclude about a difference between population. Therefore, there are no “significantly different” individuals. How the treated/control ratio is calculated, if a control group consist of many plants? What is actually null experiment standing for, if plants were younger than analyzed once?
Answer: Our poor word choice is to blame for the misunderstanding. A dot symbolizes the treated/control ratio of each measurement time point (i.e. measurement day), in other words the difference between two groups. To compute the treated/control ratios, divide the median value of all RF-EMF-exposed plants by the median value of all control plants, thus there is one single treated/control ratio per measurement time point. Null measurements are those taken immediately before the introduction of RF-EMF. Null measures demonstrate the innate disparities between two groups, which should be as small as feasible.
A more conventional data presentation (boxplots) is included in the Supplementary Materials, but it is evident that it takes up too much space and is thus unsuitable for inclusion in the manuscript itself.
We revised our manuscript to make it more understandable.
Comment 6: Why actually only ZEP and VDE genes were chosen? It seems a bit arbitrary.
Answer: Previously, we studied five stress-related genes: calmodulin, heat-shock protein HSP70B, phenylalanine ammonia lyase PAL, VDE, and ZEP with RF-EMF exposure times of up to 6 hours (as previously described in the literature), but found no difference between control and treated plants. Only ZDP and VDE might be assumed to have a slight declining trend (p < 0,1). As a result, we chose to focus our efforts on these two genes. There is no information available for the other genes with larger exposure lengths.
Comment 7: Anyway, conclusions from transcripts level only is not enough. ZEP and VDE activity can be easily assayed. Additionally, if these proteins are really affected, there should be a change in photosynthetic pigments also, at least of the xanthophyll pool.
Answer: We conducted further experiments to demonstrate the link between photosynthetic deterioration and high light stress in RF-EMF exposed plants. Furthermore, we were able to demonstrate that exposed plants had lower non-photochemical quenching (NPQ) than control plants. These findings are consistent with the decreased levels of VDE and ZEP in exposed plants.
Comment 8: I cannot agree, with presented data points, with “cyclical change” the authors describe (line 328), and, especially, the parallels built for pulse models of RNAse activity. The mechanisms for cycle occurring in minutes cannot be simply transferred into mechanisms for cycle in days.
Answer: To highlight the cyclical pattern of gene expression in exposed plants, we amended Figure 6 and added a statistical comparison of the VDE and ZEP levels with their zero levels.
In the case of VDE, there was no statistically significant difference between groups after 6 hours. However, VDE levels in the control groups were consistently and statistically significantly higher than the zero level after 12, 24, and 48 hours. In contrast, in the RF-EMF exposed plants, VDE levels dropped below zero level after 12 hours then increased above zero level after 24 hours (p < 0.05 in both cases), before dropping back to zero level after 48 hours (p > 0.05).
In the case of ZEP, a similar tendency is observed: ZEP levels after 6, 12, 24, and 48 hours in control plants were statistically significantly greater than zero level. ZEP levels in RF-EMF exposed plants were higher than zero level after 6 hours (p 0.05), equal to zero level after 12 hours (p > 0.05), higher than zero level again after 24 hours (p 0.05), and equal to zero level again after 48 hours (p > 0.05).
We agree that our comparison to the pulse models is very tentative. As a matter of fact, we lack sufficient data to speculate on the mechanism by which RF-EMF interfere with VDE and ZEP expression. The complicated pattern of VDE and ZEP expression in exposed plants might indicate the presence of an undiscovered intermediate factor that was activated by RF-EMF and controlled VDE and ZEP expression. This section has been rewritten to better demonstrate this argument.
Comment 9: Anyway, the changes in VDE/ZEP transcript level may correlate with photoperiod (which is not marked, why?).
Answer: In the experiment, light was constantly on. There was no dark period. In retrospect, the absence of dark time may have been stressful for plants. This explains the overall rise in VDE and ZEP levels in control plants.
Comment 10: Figures 6 legends says “the average transcripts level at time 0 are determined as 1”/ Why then the zero level is 0.75 and 1.25?
Answer: Because there were two Time 0 samples from two phytochambers (Control 0h and Exposed 0h, Figure 8), there are accordingly two zero levels at 0.75 and 1.25, reflecting average values from each sample. A statistical test reveals that there are no statistically significant differences between these two.
In retrospect, it would have been preferable to combine all time 0 samples into a single value, which is determined as 1. We updated our figures to reflect this.
Round 2
Reviewer 2 Report
Authors improved the manuscript. I do not have any comments or questions.
Reviewer 3 Report
I recommend acceptance.